# The Far-Infrared Absorption Spectrum of HD^16^O: Experimental Line Positions, Accurate Empirical Energy Levels, and a Recommended Line List

**DOI:** 10.3390/molecules29235508

**Published:** 2024-11-21

**Authors:** Semen N. Mikhailenko, Ekaterina V. Karlovets, Aleksandra O. Koroleva, Alain Campargue

**Affiliations:** 1V.E. Zuev Institute of Atmospheric Optics, SB, Russian Academy of Science, 1, Academician Zuev Square, 634055 Tomsk, Russia; semen@iao.ru; 2University Grenoble Alpes, CNRS, LIPhy, 38000 Grenoble, France; ekarlovets@gmail.com (E.V.K.); koral@ipfran.ru (A.O.K.); 3Department of Optics and Spectroscopy, Tomsk State University, 36, Lenin Avenue, 634050 Tomsk, Russia; 4A.V. Gaponov-Grekhov Institute of Applied Physics of the Russian Academy of Sciences, 40, Ul’yanov Street, 603950 Nizhny Novgorod, Russia

**Keywords:** water vapor, deuterated water, far infrared, rotational spectrum, deuterium, line list

## Abstract

The far-infrared absorption spectrum of monodeuterated water vapor, HD^16^O, is analyzed using three high-sensitivity absorption spectra recorded by high-resolution Fourier transform spectroscopy at the SOLEIL synchrotron facility. The gas sample was obtained using a 1:1 mixture of H_2_O and D_2_O leading to a HDO abundance close to 50%. The room temperature spectra recorded in the 50–720 cm^−1^ range cover most of the rotational band. The sensitivity of the recordings allows for lowering by three orders of magnitude the detectivity threshold of previous absorption studies in the region. Line centers are determined with a typical accuracy of 5 × 10^−5^ cm^−1^ for well-isolated lines. The combined line list of 8522 water lines is assigned to 9186 transitions of the nine stable water isotopologues (H_2_^X^O, HD^X^O, and D_2_^X^O with X = 16, 17, and 18). Regarding the HD^16^O isotopologue, a total of 2443 transitions are presently assigned while about 530 absorption transitions were available prior to our SOLEIL recordings. The comparison with the HITRAN list of HD^16^O transitions is discussed in detail. The obtained set of accurate HD^16^O transition frequencies is merged with literature sources to generate a set of 1121 accurate empirical rotation–vibration energies for the first five vibrational states (000), (010), (100), (020), and (001). The comparison to the previous dataset from an IUPAC task group illustrates a gain in the average energy accuracy by more than one order of magnitude. Based on these levels, a recommended list of transitions between the first five vibrational states is proposed for HD^16^O in the 0–4650 cm^−1^ frequency range.

## 1. Introduction

We are involved in a long-term project aiming at improving our knowledge of the spectrum of the nine stable isotopologues of water vapor in the far infrared (50–720 cm^−1^). This work relies on a series of high-quality spectra recorded during a one-week measurement campaign in October 2021, at the AILES beam line of the SOLEIL synchrotron near Paris (https://www.synchrotron-soleil.fr/en accessed on 10 April 2024). A total of twenty-one spectra of water vapor samples with a variety of isotopic compositions (natural, ^17^O-enriched, D_2_O, H_2_O:D_2_O mixture, and a mixture of the ^17^O-enriched sample with D_2_O) were recorded by high-resolution Fourier transform spectroscopy (FTS) at room temperature. The experimental conditions are listed in Table 1 where the isotopic composition of the sample injected in the cell is indicated. [In fact, due to the statistical exchange of the oxygen and hydrogen atoms between the various water molecules and to exchanges between the gas phase and water molecules adsorbed on the walls of the cell, the nine stable isotopologues (H_2_^X^O, HD^X^O, and D_2_^X^O with X = 16, 17, and 18) are present in each sample and may contribute to the recorded spectra.]

The present contribution is devoted to the analysis of the spectra #19–21 corresponding to a 1:1 mixture of H_2_O and D_2_O providing a maximum relative abundance in HDO (50%). It follows the analysis of the natural sample spectra (#1–5) [1], ^17^O-enriched spectra (#6–9) [2], and D_2_O spectra (#15–18) [3]. These studies indicated that as a result of the brightness of the synchrotron radiation and of the long pathlength (151.75 m), the sensitivity of the SOLEIL absorption spectra improves by about three orders of magnitude that of all the previous absorption studies available in the region. Our detectivity threshold corresponds typically to line intensities of 10^−25^ cm/molecule while the most sensitive previous observations were at the 10^−22^ cm/molecule level [1,2,3]. From the large sets of newly measured line positions, energy levels can be newly determined, and a general improvement in the energy level accuracy has been achieved for H_2_^16^O [1], H_2_^17^O and HD^17^O [2], D_2_^X^O (X = 16–18) [3], and H_2_^18^O and HD^18^O [4]. These previous works included systematic comparisons to the current water line lists provided by spectroscopic databases [5,6,7] and proposed a number of improvements.

The studied FIR region corresponds to the purely rotational transitions in the (000) ground vibrational state and weaker rotational transitions in the first vibrational states which are slightly populated at room temperature. It is worth noting that the lower and upper states of the observed FIR transitions are the lower states of transitions observed in all the spectral ranges. Thus, the correction of energy levels based on the studied FIR spectra propagates over the whole frequency range of the water vapor spectrum. This gives particular importance to the FIR region in the elaboration of spectroscopic databases.

Furthermore, the FIR region is of first importance for the Earth’s radiation budget. The thermal radiation mission FORUM (Far-infrared-Outgoing-Radiation Understanding and Monitoring; https://www.forum-ee9.eu/ accessed on 10 September 2024) of the European Space Agency (ESA) to be launched in 2027 will be dedicated to the “observational gap across the far-infrared (from 100 to 667 cm^−1^), never before sounded in its entirety from space”. Water vapor absorption being very strong in this region (line intensities up to 3 × 10^−18^ cm/molecule), a prerequisite for remote sensing is an accurate characterization of the water vapor spectrum including the weak lines due to the minor isotopologues as they may overlap the absorption features used to monitor other species of interest.

As mentioned above, the present work is mainly dedicated to the HD^16^O species which is the fourth most abundant isotopologues in natural water (abundance of 3.1 × 10^−4^). Although a large number of new HD^16^O transitions was identified in the analysis of the D_2_O spectra (#15–18) [3], the set of observations will be significantly enlarged using spectra #19–21 which correspond to the maximum possible HD^16^O abundance of 50%. In the next section, we present shortly the recordings, the line list retrieval, and the frequency calibration of spectra #19–21. The rovibrational assignments are presented in Section 3. Although the nine stable water isotopologues are found to contribute to the analyzed spectra, a detailed analysis will be mostly focused on the HD^16^O species. In Section 4, we propose a new set of HD^16^O empirical energy levels for the first five vibrational states: (000), (010), (100), (020), and (001) based on the HD^16^O line positions derived from spectra #19–21, from spectra #15–18 analyzed in Ref. [3] and a selection of previous works by absorption spectroscopy. The comparison to the previous set of energy levels derived ten years ago by a task group (TG) of the International Union of Pure and Applied Chemistry (IUPAC-TG) will be discussed. As the main output of the present work, an empirical line list will be generated for HD^16^O in the 0–4650 cm^−1^ region using as a basis the results of the variational calculations by Schwenke and Partridge (SP) [8,9], most of the line positions being adjusted according to the accurate empirical values of the lower and upper energy levels derived in this work.

## 2. Experiment and Line List

The unique properties of the AILES beam line at the SOLEIL synchrotron source in terms of brightness, broad-band emission, and stability make it an ideal tool for high-resolution FTS in the FIR. This applies not only to absorption lines but also to weak absorption continua, in particular the water vapor continuum [10,11,12].

### 2.1. FTS Recordings

The twenty-one FTS spectra listed in Table 1 were recorded following a mostly identical procedure which has been described in detail in Refs. [1,2,3] and will not be repeated here. Briefly, a Bruker 125 interferometer with a 6 µm mylar-composite beam splitter and a 4 K cooled Si bolometer detector were used for the recordings covering the 50–720 cm^−1^ range. The multipass absorption cell is used in a White-type configuration and has a 252 cm length. The total absorption pathlength was set to 151.75 ± 1.5 m corresponding to 60 passes. For spectra #19–21 under analysis, the cell was filled with a 1:1 mixture of H_2_O and D_2_O in order to maximize the HDO abundance at 50%. The used D_2_O sample (from Sigma-Aldrich, St. Louis, MO, USA) has a stated enrichment in deuterium larger than 99.96%. The sequence of the recordings and the corresponding experimental conditions are detailed in Table 2. The first spectrum (#19) was recorded at a pressure of 4 mbar, and then part of the sample was evacuated and a second spectrum was recorded at 0.3 mbar. The last spectrum (#21) was acquired by pumping continuously on the cell in order to measure part of the stronger lines which are saturated at higher pressure (see Figure 1). The used gauge does not allow measuring pressures in those conditions. The #21 pressure value of 10 µbar is an estimated value obtained from a comparison with variational intensities performed after the rovibrational assignment of the spectrum (see below). Spectrum #19 at 4 mbar was recorded with a spectral resolution of 0.002 cm^−1^ while, for the two others at lower pressure, the maximum spectral resolution of 0.00102 cm^−1^ was adopted. About 200 scans were co-added corresponding to an acquisition time of about 10 h at 0.001 cm^−1^ spectral resolution. The baseline fluctuations were corrected by division by a spectrum acquired at a lower resolution (0.05 cm^−1^), prior to or after each high-resolution recording (see Table 2). During the recordings, the temperature was found to vary in the 295.5 ± 0.3 K interval. An overview of spectrum #19 recorded at 4 mbar is displayed in Figure 1, which includes successive zooms illustrating the spectral congestion and the sensitivity.

The transmittances of spectra #19–21 obtained after baseline correction are provided as a Appendix A.

### 2.2. Global Experimental Line List

Let us first indicate that considering the high quality of the ab initio intensity values in the considered low energy region, the experimental accuracy required for valuable tests of the intensity calculations (1% or below) seems to be out of reach with the FTS spectra at disposal. Our uncertainties on the experimental intensities are related to the fact that (i) even at the highest resolution of the recordings (0.001 cm^−1^), the apparatus function gives a dominant contribution to the line profile; (ii) the line profile is described by a too small number of points [typically, four points at full width at half-maximum (FWHM)]; and (iii) an accurate determination of the isotopic composition of our samples is difficult. The goal of the line parameter retrieval was, thus, mainly to determine accurately the line centers and provide reasonable intensity values (which are valuable in the assignment process). From comparisons to the ab initio values, we estimate that our intensity values are accurate within 10% in the best cases (dominant isotopologues, spectrum #19 or #20, intermediate line intensity, and unblended lines). Each of the #19–21 transmittance spectra was fitted independently assuming the standard Voigt line profile as line shape (with adjusted Gaussian and Lorentzian widths) and no particular care was taken for the treatment of the apparatus function. The line parameters were obtained using a homemade multiline fitting program written in LabVIEW with DLL written in C++. Figure 2 illustrates the line profile fitting of the three spectra in a small spectral interval near 356.4 cm^−1^. The large range of the pressures of the recordings (about a factor of 400) and small noise level allowed retrieving line intensities spanning nearly five orders of magnitude. Saturated lines (transmittance at center less than a few %) were omitted from the fit when a lower pressure spectrum was available. The (obs. − calc.) residuals of the transmittance included in Figure 2 are close to the noise level [(*α_min_L*) ~ 1% root mean square (*RMS*)]. Taking into account the absorption pathlength, *L* = 151.75 m, this value corresponds to a noise equivalent absorption, *α_min_*, of 7 × 10^−7^ cm^−1^. At the highest pressure (4 mbar), the achieved noise level allows for the detection of lines with an intensity as low as 2 × 10^−25^ cm/molecule, as illustrated by the D_2_O line with an intensity of 4.0 × 10^−25^ cm/molecule observed at 356.34175 cm^−1^ in Figure 2.

The global line list obtained by combining the three individual lists counts a total of about 8600 entries and is provided as a Appendix A. Depending on the line intensity, for each line, the parameters retrieved from the spectrum corresponding to the best condition were selected. The source (#19, 20, or 21) is indicated for each line. The spectra at 10 µbar (#21), 0.3 mbar (#20), and 4 mbar (#19) were used for about 3140, 3420, and 2030 lines, respectively. The HD^16^O lines are plotted in Figure 3 with distinct symbols according to the used spectrum. The variational spectrum (SP) calculated by Tashkun using the results of Schwenke and Partridge [8,9] and available at https://spectra.iao.ru/ accessed on 10 September 2024, is plotted as background. As obvious from Figure 3, the intensity values of the strongest lines are underestimated by a factor that can be larger than 10. These intensity values were retrieved from spectrum #21 at the lowest pressure (~10 µbar). Even at this residual pressure obtained by evacuating the cell by continuously pumping during the recording, the lines with intensity larger than 10^−20^ cm/molecule remain strongly saturated leading to inaccurate intensity values. The accuracy of the corresponding line position is also affected. The position uncertainty as provided by the fit and included in the SM1 global list will be taken into account in the derivation of the energy level values (see below).

The experimental intensities of the global line list include the isotopic abundance factor which depends on the spectra. The variation in the isotopic composition is due to exchanges between the gas phase and water adsorbed in the walls of the cell which has a different isotopic composition reflecting the “history” of the cell (see Table 1). After the assignment of the spectra (see next section), it was possible to estimate the relative abundances for the nine isotopologues in each spectrum by intensity comparison with SP variational intensities [8,9]. The obtained abundance values can be found in the headings of the SM1 global list while the minimum and maximum abundances are given in Table 3. Overall, the H_2_O:HDO:D_2_O abundance ratio remains close to 1:2:1 in the three spectra #19–21 but the small abundances of the ^17^O and ^18^O isotopologues vary greatly according to the spectrum. The highest concentration of these isotopologues is found in spectrum #21 at the lowest pressure. This is probably due to the continuous desorption of ^17^O- and ^18^O-enriched water from the walls of the cell which has a larger impact at the low 10 µbar pressure of spectrum #21.

The SP variational intensities are included for comparison in the global experimental line list. For each isotopologue, SP intensities were multiplied by a factor independent of the spectrum and roughly corresponding to the maximum abundance value of the considered isotopologue (see Table 3).

The absolute frequency calibration of the global list was performed considering the line positions of spectra #20 and #21 which are not significantly affected by the self-pressure shift [13,14]. The experimental positions of about 290 H_2_^16^O lines (ν^obs^) were compared to reference values reported in Ref. [15] with accuracy better than 1 × 10^−6^ cm^−1^. The differences between the experimental line centers and the reference values were fitted as a linear function. The obtained empirical correction of the frequencies is *d*ν^corr^ = +8.5 × 10^−5^ – 7.0 × 10^−7^ν^obs^. An *RMS* deviation of 2.76 × 10^−5^ cm^−1^ was obtained for the linear fit, thus mostly determined by the experimental uncertainty on the line centers. This value gives an estimate of our accuracy on the reported positions of “good” lines. It is worth noting the consistency of the present frequency calibration with those performed in Refs. [2,3] following the same procedure. The differences between the correction laws are at most 2 × 10^−5^ cm^−1^ for the whole spectral region, confirming the stability of the experimental set up over the measurement period.

In the Appendix A global line list, the fit error on the line position determination is included. For a significant fraction of the lines, the fit uncertainty was found to be smaller than 3 × 10^−5^ cm^−1^ and is believed to underestimate the real uncertainty on the line position. For all these lines, we replaced the fit uncertainty with a value of 3 × 10^−5^ cm^−1^. Note that the position uncertainty of the weakest, highly blended lines or saturated lines can reach a value of 8 × 10^−4^ cm^−1^ in the worst cases.

## 3. Rovibrational Assignments

Over a total of more than 8600 lines, 8522 were assigned to 9186 transitions belonging to the nine stable water isotopologues. The number of transitions, maximum values of the *J* and *K* quantum numbers, and range of observations are given in Table 3 for each isotopologue. [In the following, we will use the standard normal-mode–rigid-rotor notation ($V_{1}^{\prime }V_{2}^{\prime }V_{3}^{\prime })\;J^{\prime }K_{a}^{\prime }K_{c}^{\prime }\leftarrow (V_{1}^{\prime \prime }V_{2}^{\prime \prime }V_{3}^{\prime \prime })\;J^{\prime \prime }K_{a}^{\prime \prime }K_{c}^{\prime \prime }$ to designate the transitions, where $V_{1},\;V_{2},\;and\;V_{3}$ are the vibrational quantum numbers for the symmetric stretch, bend, and asymmetric stretch modes, respectively, and *J*, *K_a_*, and *K_c_* are rigid-rotor asymmetric-top quantum numbers. The single and double primes correspond to the final (upper) and initial (lower) transition states, respectively.] After the assignment of a few impurity lines (50, 5, and 3 lines of CO_2_, HF, and DF, respectively), 48 very weak lines were left unassigned at the end of the assignment procedure.

### 3.1. H_2_^X^O (X = 16, 17, and 18)

Detailed reviews of the literature studies of vibrational–rotational spectra of H_2_^16^O, H_2_^17^O, and H_2_^18^O in our spectral region have been included in Ref. [1], Ref. [2], and Ref. [4], respectively. The SOLEIL spectra recorded with suitable isotopic enrichment (see Table 1) made it possible to significantly increase the number of observed lines of these H_2_^X^O isotopologues, especially for H_2_^18^O [4] and H_2_^17^O [2]. Since the H_2_^X^O concentrations of the presently analyzed #19–21 spectra are lower than in Refs. [1,2,4], no new H_2_^X^O transitions were detected and the number of observed lines is smaller: 1035, 356, and 406 compared to 1310 in Ref. [1], 1206 in Ref. [2], and 1150 Ref. [4] for H_2_^16^O, H_2_^17^O, and H_2_^18^O, respectively.

A systematic comparison of the present H_2_^X^O line positions (*ν^TW^*) to their values in Refs. [1,2,4] (*ν^Ref^*) gives a very good agreement with the root mean square $RMS=\sqrt{\sum_{i=1}^{N}{\left(\nu_{i}^{TW}-\nu_{i}^{Ref}\right)}^{2}/N}$ values of 15.8 × 10^−5^, 12.8 × 10^−5^, and 13.7 × 10^−5^ cm^−1^ for 917, 1773, and 1777 positions from Ref. [1], Ref. [2], and Ref. [4], respectively. A similar comparison with the W2020 line list [7] gives an *RMS* = 30.2 × 10^−5^ cm^−1^ for a total of 1797 transitions.

### 3.2. D_2_^X^O (X = 16, 17, and 18)

We have dedicated Ref. [3] to the analysis of the spectra #15–18 recorded with a highly enriched D_2_O sample. The literature review of the FIR studies of the D_2_^X^O isotopologues is included in this reference. The sensitivity of the SOLEIL spectra allowed a considerable extension of the observations for these species. For instance, more than 2057 energy levels were newly reported for D_2_^16^O. The D_2_^X^O concentrations in spectra #19–21 under study are about four times lower than in Ref. [3] (see above Table 3 and Table 2 of Ref. [3]) and only a subset of the observations of Ref. [3] is presently detected: 2152, 538, and 704 for D_2_^16^O, D_2_^17^O, and D_2_^18^O, respectively, compared to 2886, 1088, and 1169. Nevertheless, twenty-eight transitions that were not reported in Ref. [3] are presently measured. All but one corresponds to transitions between levels empirically determined in Ref. [3]. The new transitions are listed in Table 4 which includes their assignment and a comparison to the empirical positions recommended in Ref. [3]. The overall agreement is satisfactory with position differences exceeding 4 × 10^−4^ cm^−1^ for only four transitions which are close to our detection level (line intensity smaller than 4 × 10^−25^ cm/molecule).

The 23 _7 16_–22 _6 17_ pure rotational transition at 460.35106 cm^−1^ has its 23 _7 16_ upper level newly detected by absorption. From the measured position value, the corresponding term value is calculated at 3875.90790 cm^−1^. Note that our transition assignment coincides with that given by Mellau et al. [16] and Zobov et al. [17] in their analysis of emission spectra. The IUPAC-TG [18] energy value of the (000) 23 _7 16_ level (3875.90782(53) cm^−1^) relies exclusively on emission data and is found in perfect agreement with our determination by absorption.

### 3.3. HD^16^O

Although the SM1 global list includes all the HD^X^O (X = 16, 17, and 18) assignments, we will limit our detailed analysis to the HD^16^O isotopologue because the relative abundance of the HD^17^O and HD^8^O species is more than one order of magnitude larger in the spectra #10–14 which remain to be treated (see Table 1). Extended new observations are expected from these spectra and a separate contribution will be dedicated to HD^17^O and HD^18^O.

In Ref. [3] dedicated to the D_2_O species (spectra #15–18), due to exchanges with water adsorbed in the cell, the HD^16^O abundance was relatively high (up to 25% in spectrum #15) and a large number of new HD^16^O absorption lines were assigned but not discussed in details. In the following, these observations will be considered together with the present results, in particular for the derivation of the energy levels.

The literature review indicates that previous studies of the HD^16^O spectrum in the rotational range are limited: (i) the line positions of 60 transitions between 151 and 420 cm^−1^ were published by Kaupinen et al. from absorption spectrum analysis of natural abundance water sample [19]; (ii) the absorption spectra of water vapor enriched in deuterium by Johns [20], Paso and Horneman [21], and Toth [22] expanded the range of observed lines to the 50.27–719.55 cm^−1^ interval and the number of transitions to 532; and finally, (iii) our recent analyses of the SOLEIL spectra [1,2,4] have increased to 781 the number of (distinct) transitions observed by absorption, all belonging to the (000)–(000) and (010)–(010) rotational bands.

Regarding emission spectroscopy, Janca et al. reported more than 1400 transitions in the 381–720 cm^−1^ region from their high-temperature emission spectra [23]. These transitions involve levels of the eight lowest vibrational states: (000), (010), (100), (020), (001), (110), (030), and (011). Among the 1422 transitions of Ref. [23], 765 belong to the (000)–(000) and (010)–(010) rotational bands but they involve high rotational levels, and only 62 of these emission transitions are observed in absorption.

In the analysis of spectra #15–18 [3], we measured 1924 absorption transitions of HD^16^O, 1039 of them being newly reported compared to both the absorption [1,2,4,19,20,21,22] and emission [23] literature studies. A total of 2443 transitions are presently measured from spectra #19–21. They belong to the four rotational bands of the first four vibrational states: (000), (010), (100), and (020). The transitions of the (020)–(020) and (100)–(100) rotational bands are observed for the first time in absorption. The band-by-band statistics and *J_max_* and *K_a max_* values are given in Table 5. A total of 533 of these 2443 transitions are new compared to all the previously reported data, including Ref. [3]. An overview of the literature and SOLEIL observations is presented in Figure 4 where new observations are highlighted. Overall, 1572 transitions observed in the SOLEIL spectra #15–18 and #19–21 are new compared to previous studies.

The *d*ν = ν*^TW^* − ν^Ref. [3]^ differences between the present line positions and those of Ref. [3] are illustrated in Figure 5. We obtain an *RMS* value of 13.8 × 10^−5^ cm^−1^ for 1907 position differences. The position differences exceed 0.001 cm^−1^ for four lines, three of them being highly blended. The examination of the spectrum indicates that the present determinations should be preferred.

Let us now consider the position comparison to the HITRAN2020 data [5]. The source of the HITRAN line positions of HD^16^O is the variational line list elaborated by Kyuberis et al. [24]. In the considered region, all the transition frequencies were adjusted according to the IUPAC-TG empirical energy levels [25]. The *RMS* value of the *d*ν = ν*^TW^* − ν*^HITRAN2020^* differences is 57.2 × 10^−5^ cm^−1^ for 2402 positions (see Figure 5). The deviation exceeds 0.001 cm^−1^ for more than 130 positions, the largest one (−0.00815 cm^−1^) corresponds to the 19 _2 18_–19 _1 19_ pure rotational transition at 235.76381 cm^−1^. Finally, more than 40 observed transitions with upper rotation number *J*′ between 21 and 25 are missing in the HITRAN line list [5,24]. Their line intensities range between 7 × 10^−29^ and 1.4 × 10^−26^ cm/molecule (taking into account the HD^16^O natural abundance of 3.10693 × 10^−4^), which is largely higher than the HD^16^O intensity cut-off in the range under study (see below).

The complete line-by-line comparison of the present HD^16^O line positions with those of Ref. [3] and HITRAN2020 [5] is provided as a Appendix A. Some examples of significant position shifts between the SOLEIL spectra and the HITRAN2020 database are illustrated in Figure 6.

## 4. Empirical Energy Levels and a Recommended Line List for HD^16^O

The present SOLEIL line positions of HD^16^O and those of Ref. [3] were combined with previous absorption studies to determine accurate empirical values of the energy levels for the first vibrational states of HD^16^O. In the set of HD^16^O levels obtained in 2010 by an IUPAC-TG [25], the large set of emission data by Janca et al. [23] (more than 11,000 positions) played a determining role. Their declared experimental uncertainty of 1.0 × 10^−3^ cm^−1^ [23] was considered as underestimated by the IUPAC-TG (only 67.9% of the 7592 lines were validated within the declared uncertainty) [25]. A value of 0.002 cm^−1^ seems to be more reasonable for the position uncertainty of these high-temperature emission data. Since our SOLEIL positions have a more than 10 times smaller uncertainty, the emission line positions of Ref. [23] were not used for the present determination of the empirical energy levels which, thus, relies exclusively on absorption data.

The collected dataset of line positions includes about 12800 entries from 0.016 to 4368.7 cm^−1^. In addition to the present HD^16^O positions and those of Ref. [3], 36 sources involving the first eight vibrational levels—(000), (010), (100), (020), (001), (110), (030), and (011)—were selected from the literature. All the available microwave measurements [26,27,28,29,30,31,32,33,34,35,36,37,38,39,40,41,42,43,44,45] and infrared absorption measurements below 4400 cm^−1^ associated with the lowest vibrational states [19,20,21,22,46,47,48,49,50,51,52,53,54,55,56,57] were taken into account. Since all the HD^16^O transitions reported in Refs. [1,2,4] were observed under better conditions in spectra #15–18 [3] and #19–21 (present study), they were not used for the energy determination. The empirical term values of the rotation–vibration levels were recovered using the Ritz combination principle. The RITZ computer code developed by S.A. Tashkun [58,59,60] was used for this purpose. The obtained set of 1121 empirical rotation–vibration energies for the first five vibrational states (000), (010), (100), (020), and (001) is given as a separate Appendix A.

Although no new energy level is determined compared to the IUPAC-TG [25], our set of energy levels represents an important gain in terms of accuracy (see the comparison in the Appendix A). The use of about 2500 high-precision positions retrieved from the SOLEIL spectra together with the rejection of the emission data [23] leads to a significant improvement in the confidence intervals. It is illustrated in Figure 7 where the histograms of the level uncertainties are compared. For 1121 levels, the mean uncertainty is 1.41 × 10^−3^ cm^−1^ with an *RMS* of 1.91 × 10^−3^ cm^−1^ for the IUPAC-TG to be compared to 3.93 × 10^−5^ cm^−1^ and 4.97 × 10^−5^ cm^−1^, respectively, in our case. We have included in Figure 7 the histogram of the *E*^TW^–*E*^IUPAC^ energy differences. On average, the IUPAC-TG values are larger by 1.92 × 10^−4^ cm^−1^ and the standard deviation (1.11 × 10^−3^ cm^−1^) is consistent with the IUPAC-TG error bars. The systematic overestimation of the IUPAC-TG energy values is illustrated in Figure 8 where the energy differences are plotted versus the energy of the rotational levels of the (000), (010), and (001) vibrational states. This figure shows that the accuracy improvement concerns not only the four rotational bands observed in the SOLEIL spectra (Table 5) but also the (001)–(001) band for which no transitions were observed in the SOLEIL spectra.

According to Figure 8, the overestimation of the IUPAC-TG energies is negligible at low rotational energy and increases mostly linearly with the energy [up to 5 × 10^−4^ cm^−1^ for rotational energies around 3000 cm^−1^ in the (000) ground state]. The deviations have similar amplitude for the different vibrational levels which might indicate that the energy differences in the ground state are propagated in the excited levels. We tried to trace the origin of this observation by considering a possible calibration error of the emission spectra of Janca et al. [23] used by the IUPAC-TG [25]. The transition frequencies reported by Janca et al. were calculated using our energy levels and compared to their original values but no clear systematic trends could be evidenced. Note that the observed deviations are clear but well below the uncertainty of Janca et al.’s positions (~10^−3^ cm^−1^).

For ten levels (not considered in the above comparison and excluded from the Appendix A), our derived energy value differs from the IUPAC-TG value by more than 0.06 cm^−1^. None of these problematic levels is involved in the FIR transitions observed in the SOLEIL spectra. Thus, our energy values rely exclusively on one of the thirty-six sources of absorption line positions that we selected in the literature, more specifically from Refs. [46,48,53]. While all the IUPAC-TG energies were determined from several (from 3 to 9) emission line positions reported by Janca et al. [23], all but one of our values rely on a single transition from Refs. [46,48,53], as detailed in the following:

(i)The (010) 20 _0 20_ and (010) 20 _1 20_ energies (*dE* = −0.096 cm^−1^) were obtained from the 1643.8755 cm^−1^ line position assigned by Toth to the 20 _0 20_–19 _1 19_ and 20 _1 20_–19 _0 19_ transitions of the ν_2_ band [53],(ii)The (100) 13 _7 7_ and (100) 15 _2 14_ levels with the *dE* values of −0.097 and −0.061 cm^−1^, respectively, were determined on the basis of the results of Papineau et al. [46]. The (100) 13 _7 7_ energy relies on the 3078.740 cm^−1^ wavenumber assigned to the 13 _7 7_–12_6 6_ transition of the ν_1_ band. The (100) 15 _2 14_ energy was obtained from the 2904.200 and 2904.655 cm^−1^ wavenumbers assigned to the 15 _2 14_–14 _2 13_and 15 _2 14_–14_1 13_ transitions of the same ν_1_ band,(iii)(The 9 _9 1_, 9 _9 0_, 10 _8 3_, 10 _9 2_, 10 _9 1_, and 15 _2 14_ levels of the (001) vibrational states have *dE* values of −0.097, −0.097, 0.081, −0.140, −0.140 and 0.266 cm^−1^, respectively. Our energy value relies on line positions of ν_3_ transitions given by Toth and Brault [48]: 9 _9 1_–8 _8 0_ and 9 _9 0_–8 _8 1_ at 4007.7146 cm^−1^, 10 _8 3_–9 _7 2_ at 4012.5627 cm^−1^, 10 _9 2_–9 _8 1_ and 10 _9 1_–9 _8 2_ at 4021.6755 cm^−1^, 15 _2 14_–14 _2 13_ at 3895.4251 cm^−1^.

In order to resolve these conflictive situations, we performed a comparison of the empirical energy (*E*^emp^) (obtained in this work and given by the IUPAC-TG) with the corresponding SP variational values (*E*^SP^) from Ref. [8]. Indeed, it is well known that the *dE* = *E*^emp^–*E*^SP^ energy differences show a smooth dependency on the rotational numbers *J* and *K_a_* (see, for example, Figure 6 in Ref. [16] or Figure 7 in Ref. [61]). For example, the smooth *J* dependence of the *dE* energy difference in the (010) *E*(*J K_a_* = 0 *K_c_* = *J*) series allows for predicting the energy of the (010) 20 _0 20_ level within typically 5 × 10^−3^ cm^−1^ and thus discriminating the correct value between our energy and the IUPAC-TG value which differ by nearly 0.1 cm^−1^. In this case and for the nine other levels, the IUPAC-TG energy values obtained from emission transitions [23] were confirmed and thus found preferable to those obtained from the absorption transitions of Refs. [46,48,53], indicating a probable erroneous assignment in those references.

The overview of the 4451 HD^16^O transitions provided by the HITRAN2020 database in the 50–720 cm^−1^ range is presented in Figure 9. The HITRAN intensity cut-off (including the HD^16^O natural abundance factor of 3.10693 × 10^−4^) varies from 1.8 × 10^−30^ cm/molecule at 50 cm^−1^ to 16 × 10^−30^ cm/molecule at 720 cm^−1^. The HITRAN maximum value of the rotational quantum number is *J* = 20 but in the present SOLEIL spectra, transitions with *J* values up to 25 were assigned and have intensities larger than the HITRAN intensity cut-off. In Figure 9, we have superimposed to the HITRAN list the SP variational transitions with *J* > 20 [8,9]. Overall, about 400 *J* > 20 transitions are predicted with an intensity larger than 1 × 10^−30^ cm/molecule. About 160 have an intensity larger than the HITRAN cut-off (the maximum intensity value of these missing transitions is above 1 × 10^−26^ cm/molecule).

A recommended absorption line list at 296 K for the different bands of HD^16^O involving the (000), (010), (020), (001), and (100) vibration states is provided as a Appendix A. This list uses as a basis the SP variational line list calculated by Tashkun using the VTET program of Schwenke [62]. The intensity cut-off was fixed to 1 × 10^−27^ cm/molecule for pure HD^16^O at 296 K (3 × 10^−31^ cm/molecule if the abundance factor is included). The line list includes about 31000 transitions belonging to a total of fourteen bands and spans the 0–4650 cm^−1^ range. For all the transitions involving lower and upper levels with known empirical energy values, the variational frequency has been substituted by its empirical value using our energy values when available and the IUPAC-TG values otherwise. For a small fraction of transitions, empirical energy values are missing and the SP variational frequency was kept unchanged. The overview of the recommended list is given on the upper panel of Figure 10 where transitions with the empirical and variational frequencies are presented with different symbols. Note that while in our region corresponding to the rotation bands (0–700 cm^−1^), it has been possible to empirically correct most of the line positions using our energy values; at higher frequencies, a significant fraction of the transitions have their frequencies relying on the IUPAC-TG energy levels.

## 5. Concluding Remarks

Three long-pass FTS spectra recorded in the far infrared at the SOLEIL synchrotron have been analyzed to extend the set of HD^16^O lines measured by absorption. Overall, the previous [1,2,3,4] and present analyses of different SOLEIL spectra have increased the number of absorption line positions measured between 50 and 720 cm^−1^ from about 530 available in the literature to more than 2350. For the first time, rotational absorption lines in the (010), (100), and (020) excited states were detected. The typical line position accuracy (better than 10^−4^ cm^−1^) has allowed for the reduction in the error bars of the energy levels of the first five vibrational states of HD^16^O. Compared to the most relevant previous set elaborated by an IUPAC task group fifteen years ago [25], the average energy uncertainty is reduced from 1.4 × 10^−3^ cm^−1^ to about 4 × 10^−5^ cm^−1^. As the considered states are the lowest states of most of the rovibrational transitions, this improvement will impact the accuracy of the line positions in all the frequency regions of the HD^16^O spectrum.

As the main output of the present work, a recommended line list is provided for the fourteen HD^16^O bands involving the first five vibrational levels. This list covering the 0–4650 cm^−1^ frequency range is proposed for improving the line list of natural water vapor in the far-infrared region, of particular importance for atmospheric applications.

## Figures and Tables

**Figure 1 molecules-29-05508-f001:**
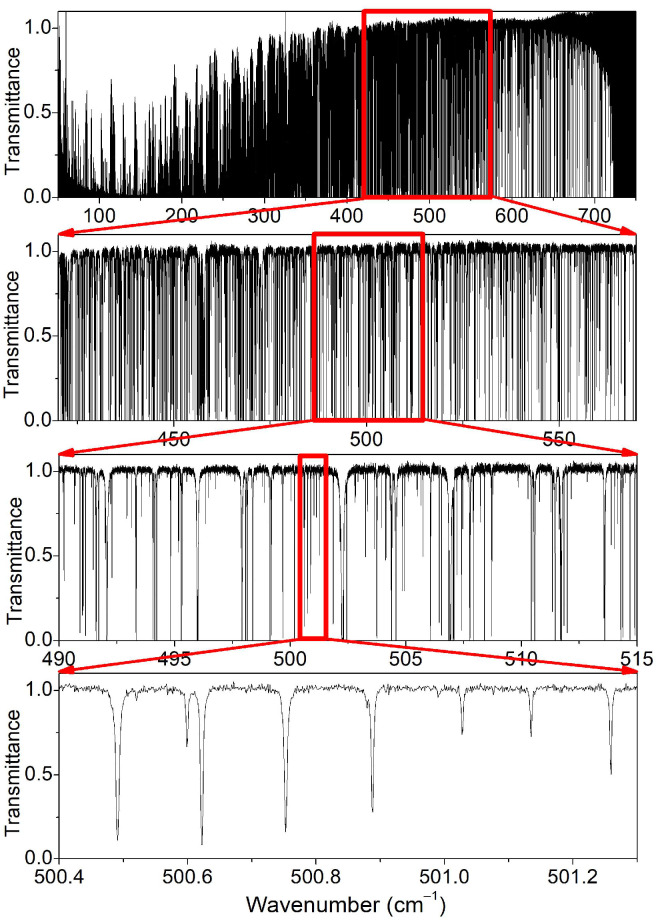
Successive zooms of the FTS spectrum #19 of deuterated water recorded at the SOLEIL synchrotron with a pressure of 4 mbar between 50 and 700 cm^−1^.

**Figure 2 molecules-29-05508-f002:**
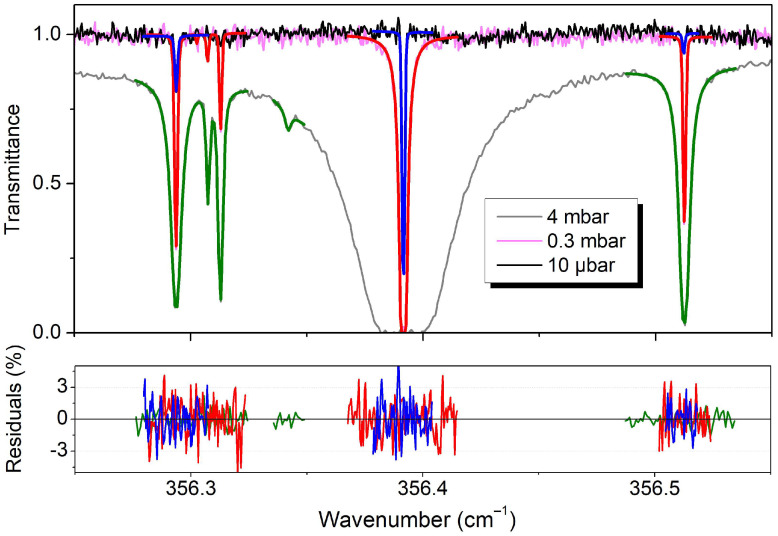
Line parameter retrieval from the FTS spectra #19–21 of deuterated water near 356.4 cm^−1^. The line profile fit was performed in narrow spectral intervals around the lines that were not too saturated. **Upper panel***:* Recorded spectra at about 10 µbar, 0.3 mbar, and 4 mbar (#21, #20, and #19, respectively) with corresponding best-fit spectra (blue, red, and green, respectively). **Lower panel***:* Corresponding (obs. − calc.) residuals in %.

**Figure 3 molecules-29-05508-f003:**
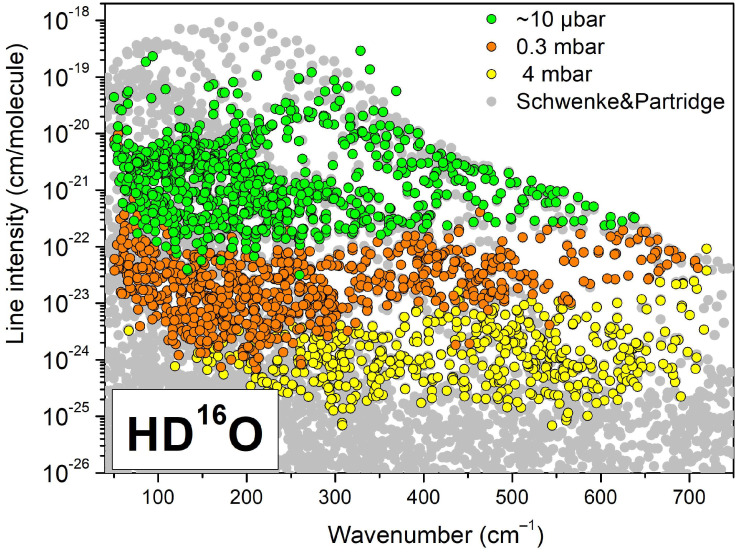
Overview of the HD^16^O lines retrieved from spectra #19–21 of deuterated water between 50 and 720 cm^−1^. The global experimental line list was obtained by combining the lists at about 10 µbar (#21), 0.3 mbar (#20), and 4 mbar (#19) (green, orange, and yellow dots, respectively). Note that the strongest lines retrieved from the lowest pressure spectrum are measured with strongly underestimated intensities (see text). The gray dots correspond to the SP variational list [8,9].

**Figure 4 molecules-29-05508-f004:**
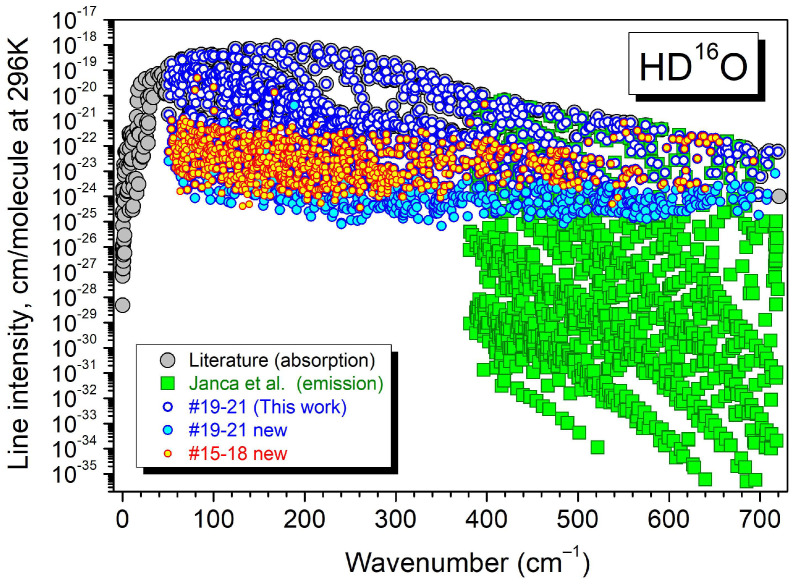
Overview of the HD^16^O transitions observed by absorption and by emission below 720 cm^−1^ [23]. The line intensities from Schwenke and Partridge [8,9] were attached to the transition wavenumbers.

**Figure 5 molecules-29-05508-f005:**
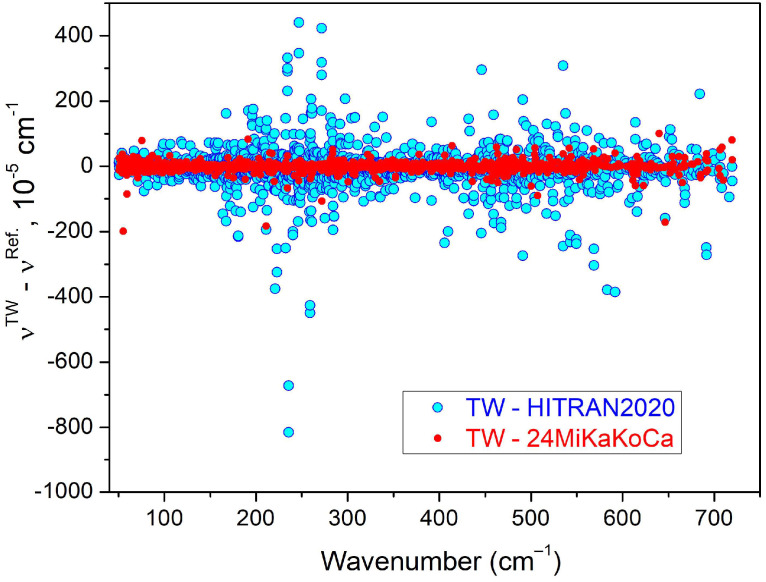
Differences between the HD^16^O line positions measured from the SOLEIL spectra in the present study and in Ref. [3] (24MiKaKoCa) (red dots) and differences between the present and HITRAN2020 values [5] (cyan dots).

**Figure 6 molecules-29-05508-f006:**
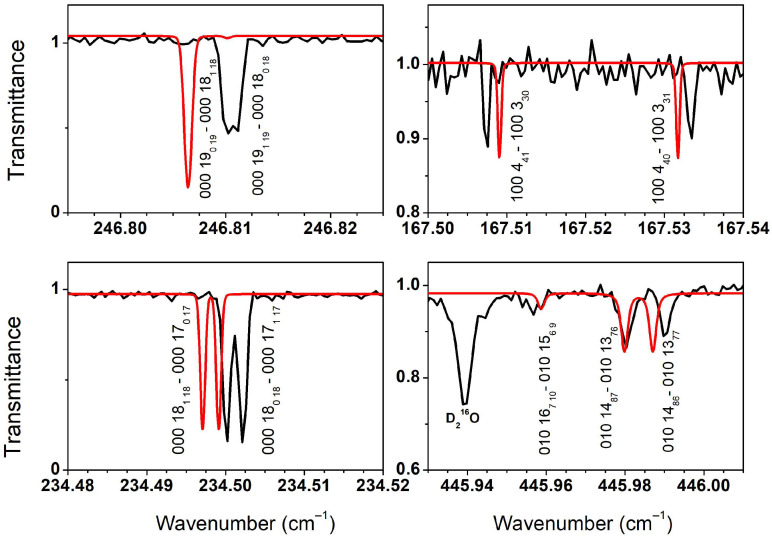
Comparison between the FTS spectrum (black line) of HD^16^O transitions to the spectra simulations (red line) based on the HITRAN2020 database in four spectral intervals showing significant differences.

**Figure 7 molecules-29-05508-f007:**
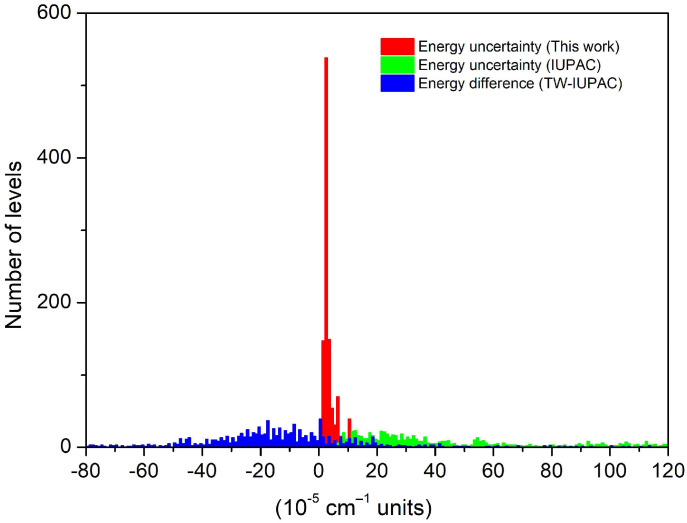
Histograms of the uncertainties of the present and IUPAC-TG [25] energy levels of HD^16^O (red and green, respectively). The blue histogram corresponds to the differences in the energy values obtained in this work (TW) and those recommended by the IUPAC-TG.

**Figure 8 molecules-29-05508-f008:**
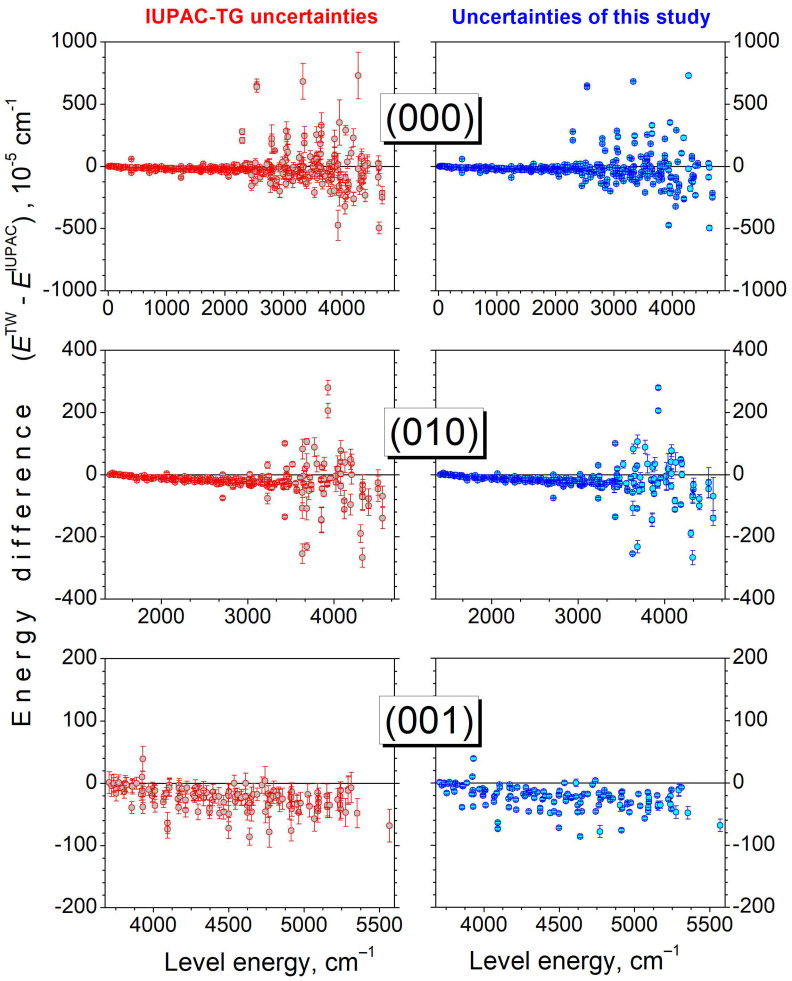
Energy differences, *E*^TW^–*E*^IUPAC^, of the energy levels of the ground, (010), and (001) vibrational states of HD^16^O determined in this work (TW) and recommended by the IUPAC-TG [25].

**Figure 9 molecules-29-05508-f009:**
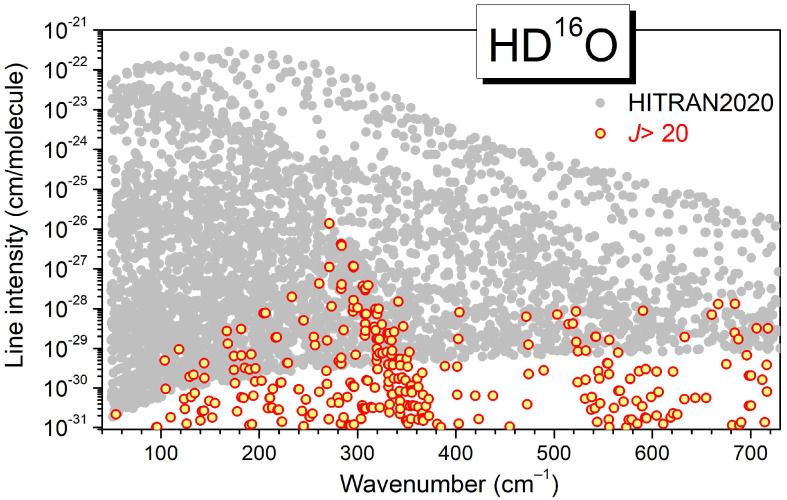
Overview of the HD^16^O line list provided in the HITRAN2020 database (gray dots) [5] and of the lines with *J* > 20 predicted by Schwenke and Partridge (red circles) [8,9]. The HITRAN list is limited to *J* ≤ 20 transitions.

**Figure 10 molecules-29-05508-f010:**
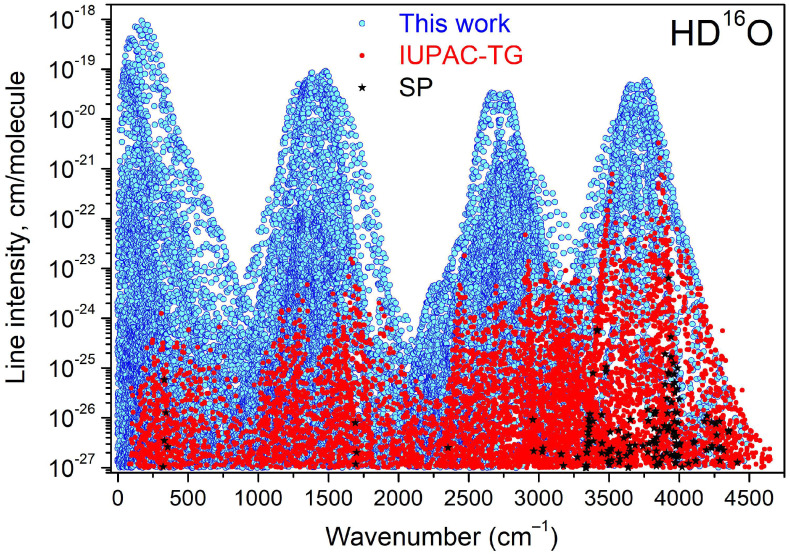
Overview of the HD^16^O recommended line lists based on the variational calculations by Schwenke and Partridge [8,9]. Except those corresponding to the black stars, the transition frequencies have been empirically corrected, using either presently determined or IUPAC-TG energy levels (cyan and red symbols, respectively).

**Table 1 molecules-29-05508-t001:** Experimental conditions of the room temperature FTS spectra of the various isotopic samples of water vapor recorded at the SOLEIL synchrotron. The absorption pathlength was 151.75 m.

Label	Sample *^a^*	Max. Pressure (mbar)	Nb of P *^b^*	Ref.
#1–5	Natural	6.7	5	[1]
#6–9	41% H_2_^17^O, 27% H_2_^18^O, 22% H_2_^16^O	3.85	4	[2]
#10–14	50% D_2_O 20.5% H_2_^17^O 13.5% H_2_^18^O 11% H_2_^16^O	8.0	5	
#15–18	D_2_O	8.0	4	[3]
**#19**–**21**	**50% H_2_^16^O** **50% D_2_O**	**4.0**	**3**	**This work**

Notes*: ^a^* rough isotopic composition of the water sample injected in the cell. *^b^* number of different pressure values used for the recordings.

**Table 2 molecules-29-05508-t002:** Experimental conditions of the three FTS spectra of HDO under analysis.

Label	Recording	Pressure	Resolution, cm^−1^	Number of Scans
**#19**	**Sample**	**≈4 mbar**	**0.002**	**160**
	Baseline	Pumping on the cell	0.05	200
**#20**	**Sample**	**≈0.3 mbar**	**0.001**	**100**
	Baseline	Pumping on the cell	0.05	200
**#21**	**Sample**	**≈10 µbar *^a^***	**0.001**	**160**
	Baseline	Pumping on the cell	0.05	200

*^a^* the used pressure gauge does not allow measuring so low pressure value and the given 10 µbar value is an approximate value obtained by intensity comparison with calculations.

**Table 3 molecules-29-05508-t003:** Statistics of assigned water transitions.

Molecule	Abundance, Min–Max, in %	Factor* ^a^* (%)	Number of Transitions *^b^*	*J* _max_	*K_a_* _max_ * ^c^ *	Region, cm^−1^
H_2_^16^O	24.650–24.910	25.0.0	1035	21	13	51.434–718.657
H_2_^17^O	0.025–0.140	0.15	356	16	11	53.509–667.815
H_2_^18^O	0.070–0.210	0.2	406	17	11	53.569–702.589
HD^16^O	49.250–49.785	50	2443	25	14	50.277–719.550
HD^17^O	0.045–0.320	0.3	701	19	11	50.134–631.440
HD^18^O	0.170–0.430	0.45	851	20	12	52.187–654.961
D_2_^16^O	24.330–25.865	25	2152	29	17	50.210–690.430
D_2_^17^O	0.025–0.200	0.52	538	21	14	50.425–490.178
D_2_^18^O	0.070–0.470	0.2	704	23	14	51.538–516.600

*^a^ *the calculated intensity values included in the global list were obtained by multiplying the SP variational intensity of the pure species by this factor. Note that the sum of these factors is larger than 100%. *^b^* number of assigned transitions. *^c^ J* is the rotational angular momentum quantum number and *K_a_* corresponds to the projection of the angular momentum onto the *a* axis.

**Table 4 molecules-29-05508-t004:** Comparison of newly observed D_2_^X^O (X = 16, 17, and 18) absorption line positions with their empirically calculated values from Ref. [3].

Position	*dF*	Int_SP	Iso	*V* *′*	*J* *′*	*K_a_* *′*	*K_c_* *′*	*V″*	*J″*	*Ka″*	*Kc″*	Pos_calc	*dν*
50.21024	21	2.129 × 10^−23^	D_2_^16^O	010	3	3	0	010	3	2	1	50.21043	−19
124.23846	3	1.338 × 10^−23^		010	7	7	1	010	7	6	2	124.23852	−6
124.25938	13	7.880 × 10^−25^		000	20	6	15	000	20	5	16	124.25936	2
124.31025	12	2.093 × 10^−24^		010	14	3	12	010	14	2	13	124.31029	−4
140.98902	32	2.371 × 10^−23^		000	15	9	6	000	15	8	7	140.98887	15
167.54655	22	2.078 × 10^−25^		010	12	10	3	010	12	9	4	167.54651	4
212.70450	30	3.205 × 10^−25^		020	9	5	4	020	8	4	5	212.70449	1
231.59471	3	5.643 × 10^−25^		020	9	6	4	020	8	5	3	231.59484	−13
252.97401	8	2.135 × 10^−25^		020	8	8	1	020	7	7	0	252.97418	−17
261.54655	6	2.118 × 10^−25^		000	27	0	27	000	26	1	26	261.54785	−130
261.78073	6	2.006 × 10^−25^		000	26	2	25	000	25	1	24	261.78130	−57
262.52074	50	8.240 × 10^−26^		000	13	8	6	000	14	3	11	262.52090	−16
265.59588	10	1.397 × 10^−25^		020	9	8	1	020	8	7	2	265.59627	−39
328.10894	10	1.818 × 10^−24^		010	12	11	2	010	11	10	1	328.10904	−10
365.73560	12	1.967 × 10^−25^		010	15	11	4	010	14	10	5	365.73541	19
460.35106	81	5.608 × 10^−26^		000	23	7	16	000	22	6	17		
462.29782	79	5.180 × 10^−26^		000	17	11	7	000	17	8	10	462.29779	2
544.67069	22	1.538 × 10^−24^		000	10	10	1	000	9	7	2	544.67053	16
656.61455	38	1.844 × 10^−25^		000	13	12	1	000	12	9	4	656.61476	−21
683.17223	47	1.504 × 10^−25^		000	15	12	3	000	14	9	6	683.17338	−115
62.28061	35	5.340 × 10^−23^	D_2_^17^O	000	4	4	1	000	4	3	2	62.28059	2
78.61542	9	6.384 × 10^−23^		000	9	5	5	000	9	4	6	78.61536	6
99.78119	15	9.088 × 10^−25^		000	14	5	9	000	13	6	8	99.78101	18
96.21984	14	1.755 × 10^−22^	D_2_^18^O	000	9	2	7	000	8	3	6	96.22005	−21
103.97704	17	2.675 × 10^−25^		010	6	6	1	010	6	5	2	103.97646	58
177.97477	6	9.695 × 10^−24^		010	6	6	0	010	5	5	1	177.97499	−22
285.06436	46	2.464 × 10^−25^		010	11	9	2	010	10	8	3	285.06453	−17
312.95597	21	2.174 × 10^−25^		010	11	5	7	010	10	2	8	312.95560	37

Notes: Position—measured line position (cm^−1^); *dF*—position uncertainty (10^−5^ cm^−1^); Int_SP—variational [8,9] line intensity (cm/molecule) multiplied by the maximum value of the isotopic abundances (D_2_^16^O—0.25; D_2_^17^O—0.002; D_2_^18^O—0.005); Iso—isotopologue; *V*′*J*′*Ka*′*Kc*′—vibration and rotation numbers of the upper state; *V″J″Ka″Kc″*—vibration and rotation numbers of the lower state; Pos_calc—calculated line position (recommended line lists) of Ref. [3]; *dν*—position difference *ν^TW^*—*ν*^Ref. [3]^ (10^−5^ cm^−1^).

**Table 5 molecules-29-05508-t005:** General information on the HD^16^O transitions assigned in the three analyzed spectra between 50 and 720 cm^−1^.

Band	*NT* ^ *a* ^	*J_max_*	*K_a max_*	Region, cm^−1^
(000)–(000)	1778	25	14	50.27–719.55
(010)–(010)	617	18	11	50.73–698.49
(020)–(020)	18	9	6	145.53–310.23
(100)–(100)	30	9	8	96.60–350.15
Total	2443	25	14	50.27–719.55

*^a^* number of assigned transitions.

## Data Availability

The data that support the findings of this study are available within the article and its Appendix A.

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
