# Peer review of "The Far-Infrared Absorption Spectrum of HD16O: Experimental Line Positions, Accurate Empirical Energy Levels, and a Recommended Line List"

_molecules, 2024, doi:10.3390/molecules29235508_

Round 1

Reviewer 1 Report

Comments and Suggestions for Authors

The paper presents the far infrared absorption spectrum of HD16O. This work is a continuation and deepening of the authors’ long-term and systematic research. The research team has a solid foundation in the spectroscopy of water vapor molecules and has made significant contributions to the field. In this paper, the authors have greatly enhanced the spectral line positions, line strengths, and energy levels of water vapor isotopes, deepening our understanding of water vapor molecular spectroscopy. Undoubtedly, this paper is worthy of publication.

One minor issue needs to be addressed. The authors mention, “The last spectrum (#21) was acquired by pumping continuously on the cell in order to measure part of the stronger lines which are saturated at higher pressure (see Fig. 1). The pressure value of #21, which is 10 μbar, is a rough estimation.” In the continuous pumping process, what measures were taken to maintain the gas pressure in the absorption cell at a relatively stable level? Additionally, how long was the pumping performed to achieve a dynamic equilibrium between the gas extraction and the release from the absorption cell? Given that the pressure inside the absorption cell cannot be precisely determined, what method was employed to accurately determine the line strengths of those strong lines? It would be helpful if the authors could provide some explanations regarding the determination of line strengths under such low and uncertain pressure conditions.

Author Response

We thank reviewer 1 for his/her useful comments which were taken into account as detailed in the attached file.

Reviewer 2 Report

Comments and Suggestions for Authors

The paper presents far infrared absorption spectra of monodeuterated water vapor, HD16O, obtained by a 1:1 mixture of H2O and D2O. This is the forth work written by the authors on a series of similar samples measured exploiting the unique features available at the beamline SOLEIL. The use of this set up allows the lowering of the detectivity threshold of previous absorption studies in the region. Moreover they provide a list of lines and their underlying transitions; comparison with available database are also reported.

I appreciate the technical details of the paper, but I think that in order to be published the authors should describe in the introduction section the reasons for performing this type of study and the potential importance of the reported results. Moreover the difference between previous studies and the present one should be clearly stated.

Some points should be addressed too, as detailled in the following.

-The spectra of all three samples should be displayed, even if in the SM

-Table 1. Please modify the pressure unit of measurement

-Table 2. The authors use an indicative value for the pressure during measurement of sample 21, why don’t they use an upper limit, i.e. the pressure is less than the minimum measured pressure?

- Section 2.2. The authors estimate that the intensity values are accurate within 10% in the best cases. How did they perform such evaluation?

-Section 2.2. The authors evaluate a noise equivalent absorption of the order of 7×10-7 cm-1 and a detectivity threshold of about 2×10-25 cm/molecule for the line intensities measured at the highest pressure. How did they estimate the detectivity threshold?

-Figure 2 reports three best fit lines, green blue and red lines, but it is not clear how they are obtained and what are the main differences. I also suggest to show in the SM more lines fits, choosing among the more significative frequency ranges.

-Table 3 J and K should be defined in the caption

-Authors contribution is missing.

Author Response

We thank reviewer 2 for his/her useful comments which were taken into account as detailed in the attached file.

Round 2

Reviewer 2 Report

Comments and Suggestions for Authors

The authors addressed all my comments. I only suggest to explain in the text how they estimated the detectivity threshold.
